# Pharmacophore mapping approach to find anti-cancer phytochemicals with metformin-like activities against transforming growth factor (TGF)-beta receptor I kinase: An *in silico* study

**Rumman Reza[1], Niaz Morshed[1], Md. Nazmus Samdani[1], Md. Selim Reza[2]\***

**1** Department of Pharmacy, University of Dhaka, Dhaka, Bangladesh, **2** Department of Pharmaceutical Technology, University of Dhaka, Dhaka, Bangladesh

\* selimreza@du.ac.bd

## Abstract

The most frequently prescribed first-line treatment for type II diabetes mellitus is metformin. Recent reports asserted that this diabetes medication can also shield users from cancer. Metformin induces cell cycle arrest in cancer cells. However, the exact mechanism by which this occurs in the cancer system is yet to be elucidated. Here, we investigated the impact of metformin on cell cycle arrest in cancer cells utilizing transforming growth factor (TGF)-beta pathway. TGF-ß pathway has significant effect on cell progression and growth. In order to gain an insight on the underlying molecular mechanism of metformin's effect on TGF beta receptor 1 kinase, molecular docking was performed. Metformin was predicted to interact with transforming growth factor (TGF)-beta receptor I kinase based on molecular docking and molecular dynamics simulations. Furthermore, pharmacophore was generated for metformin-TGF-ßR1 complex to hunt for novel compounds having similar pharmacophore as metformin with enhanced anti-cancer potentials. Virtual screening with 29,000 natural compounds from NPASS database was conducted separately for the generated pharmacophores in Ligandscout® software. Pharmacophore mapping showed 60 lead compounds for metformin-TGF-ßR1 complex. Molecular docking, molecular dynamics simulation for 100 ns and ADMET analysis were performed on these compounds. Compounds with CID 72473, 10316977 and 45140078 showed promising binding affinities and formed stable complexes during dynamics simulation with aforementioned protein and thus have potentiality to be developed into anti-cancer medicaments.

## Introduction

According to a research by the World Health Organization, around 10 million people died from cancer in the year 2022 [1]. Cancer is listed as the second most common cause of fatality in 91 nations and the third most common cause of death in 22 additional countries as per a

**Funding:** The author(s) received no specific funding for this work.

**Competing interests:** The authors have declared that no competing interests exist.

**Abbreviations:** ADMET, Absorption, distribution, metabolism, excretion, and toxicity; CADD, Computer-aided drug development design; MDS, Molecular dynamics simulation; Rg, Radius of gyration; RMSD, Root mean square deviation; RMSF, Root mean square fluctuation; RO5, Lipinski's Rule of Five; SASA, Solvent-accessible surface area; TGFß, Transforming Growth Factor-ß.

survey conducted by WHO for patients under the age of 70 [2]. There may be 29.4 million new cases of cancer, according to estimates by the year 2040 [1]. The alarming morbidity and mortality rates for the non-communicable disease known as cancer must be decreased in order to raise the average life expectancy worldwide. Chemotherapy, surgical removal, radiotherapy, or a combination of these treatments are all possible therapeutic choices; nonetheless, relapse and recurrence are possible, and the outcome may not be positive [3]. Additional efficient therapeutic approaches are required because the available therapy alternatives are insufficient [4].

Transforming growth factor beta (TGF-ß) is a part of a family of dimeric polypeptide growth factors that also contains bone morphogenic proteins (BMPs) and activins [5]. Almost every cell in the body produces TGF-ß and the receptors that bind to it. TGF-ß plays important roles in embryonic development, angiogenesis, wound healing, and cell proliferation and differentiation. The emergence of cancer may also be influenced by slight shifts in TGF-ß signaling [5]. These varied effects depend on the tissue and malignancy [6]. A potential area of research that may result in novel cancer prevention and treatment strategies is the identification and understanding of anomalies in the TGF-ß signaling pathway in diverse malignancies [7]. Both cell-surface serine/threonine kinase receptors, type II (TßRII) and type I (TßRI), are required for the TGFß-1 signaling cascade [8].

TGFß-I usually binds to TßRII, which then draws TßRI into a heteromeric complex of TßRI-TßRII. When this complex is formed, TßRI is phosphorylated. This leads to receptor-regulated Smads (R-Smads, such as Smad2, Smad3, etc.) being phosphorylated. The Smads also bind a co-Smad (Smad4) [8]. Target genes are subsequently controlled by the translocation and accumulation of the R-Smad/co-Smad complexes in the nucleus [9]. TGF-ß induce TßRII to dimerize from its resting monomeric form, supporting the notion that receptor dimerization is necessary for receptor activation. It has been demonstrated that metformin blocks TGF-ßI-induced EMT, which is essential for the development of cancer and organ fibrosis. Research has shown that metformin may help cancer patients have better prognoses and lower their chance of developing cancer. These results provide credence to the hypothesis that metformin inhibits the growth of malignant tumors and organ fibrosis by blocking the TGF-ßI activated pathway [10].

The first-line oral medication for the treatment of type 2 diabetes is metformin [11]; it is affordable and has a strong therapeutic impact [12]. According to studies, persons with diabetes who take metformin have a decreased incidence of cancer [12–14]. Numerous research has suggested that metformin may play a role in the treatment of cancer [12, 13, 15–17]. Recent studies looked into the potential anticancer effects of metformin in non-diabetic cancer patients with lung, breast, and prostate cancer [12–14]. These studies have examined the possible impacts of metformin in terms of cancer therapy and prevention.

Metformin may be clinically beneficial in the treatment of gynecologic malignancies, according to evidence [18]. Combining existing anticancer medications with metformin may improve their effectiveness and reduce negative side effects [16]. Metformin has been shown to have anticancer effects in both in vitro and in vivo models of multiple cancer types, according to growing data [19]. However, the underlying molecular mechanisms that are responsible for the anti-cancer activities of metformin in cancer is yet to be elucidated. In previous studies, it has been shown that metformin blocks type II TGF-β1 receptor dimerization when the receptor is exposed to TGF-β1 [20]. The dimerization step is particularly important for downstream signal transduction. Xiao et al., reported that metformin reduces TGFß-1 downstream signaling through an AMPK-independent mechanism [20]. They also found that metformin reduced the percentage of dimers induced by TGF-β1 in a dose-dependent way. Thus,

metformin can possibly block TGF-ß signaling pathway and have therapeutic potentials to be used in diseases where TGF-ß signaling hyper-functionality occurs.

TGF-ß is implicated in many other disorders besides fibrosis and malignancies [21]. Therefore, it has become desirable for medication development to target the TGF-signaling pathway. There are currently three therapeutic approaches for the TGF- family: 1) translational inhibition using antisense oligonucleotides, 2) ligand-receptor interaction inhibition utilizing ligand traps and anti-receptor monoclonal antibodies, and 3) receptor-mediated signaling cascade inhibition using inhibitors and aptamers of TGF-ß receptor kinases. The limited capacity of antisense oligonucleotides and monoclonal antibodies to access the targeted tissue, for instance, is one of the specific difficulties with these techniques that restricts their application. Metformin, in contrast, is a small molecule drug that can quickly enter the desired tissue. It is possible for other kinases to be cross-inhibited by TGF-ß receptor kinase inhibitors, which results in undesirable side effects [22]. On the other hand, during years of use, metformin has proven to be secure and to cause fewer negative effects. In addition, metformin has positive benefits that go beyond just targeting TGF-ßI, and based on the way that metformin and TGF-ßI interact, new drugs can be created to more effectively and specifically target TGF-ßI.

There are efforts to develop a structurally optimal TGFBR1 small-molecule inhibitor which will lower kinase activity more effectively than those already developed. A small-molecule inhibitor called SD-208 has recently been utilized to treat various cancers in a number of animal experiments. Another example of novel transforming growth factor beta receptor I kinase inhibitor is galunisertib (LY2157299) [23]. In a separate clinical trial, the combination of galunisertib and another kinase inhibitor called Sorafenib has been tested [24]. Activity of TGFBR1 inhibitor, capecitabine (LY3200882) on pretreated metastatic colorectal cancer is being investigated in ongoing trials [25]. The kinase inhibitor regorafenib in combination to humanized antibody PF-03446962 has been examined against TGFBR1 protein in eleven pretreated colorectal cancer patients under Phase 1 clinical trial [26, 27].

In the present study, the interaction between metformin and TGF-ß receptor 1 kinase has been studied to understand the inhibitory effect of metformin on TGF-ß receptor 1 and 2 dimerization that leads to halting of the signaling pathway. Furthermore, pharmacophore mapping technique has been utilized to search for novel compounds from 29,000 natural molecules that can mimic the action of metformin on TGF-ß receptor 1 kinase. The entire workflow used in the current study have been depicted in Fig 1. The figure shows subsequent steps and methodologies adopted in this research work.

## Materials and methods

### Investigation of transforming growth factor-beta (TGF-β) receptor type 1 (TGFBR1) mRNA expression in different cancers

The most common malignancies in terms of death rates are lung, colon and rectum, liver, stomach, and breast cancers, respectively, as per the World Health Organization Report 2020 [1]. In terms of new cancer cases, the most common cancer kinds in 2020 are breast, lung, colon, and rectum, prostate, non-melanoma skin, and stomach. The TGFBR1 mRNA expression in breast, lung, colon, and rectum cancer cells, as well as liver cancer cells, were examined using the UALCAN server. Additionally, data on TGFBR1 mRNA expression in normal tissues was gathered from the UALCAN web server. TGFBR1 expression in cancer tissues was compared to healthy tissue counterparts using the Cancer Genome Atlas (TCGA) dataset included in the UALCAN.

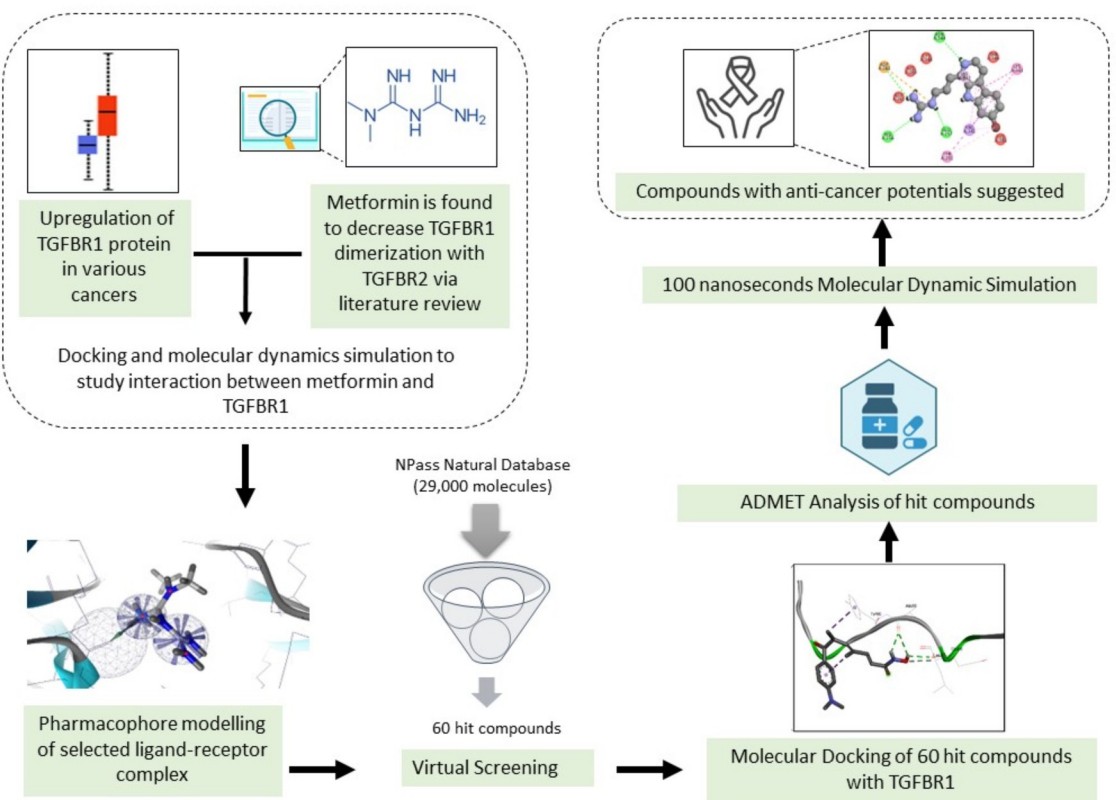

**Fig 1. A flowchart depicting the workflow followed in the current study.** The graphic depicts the succeeding processes and empirical approaches employed in the research.

## Preparation of target protein

The crystal structure of TGF-ß receptor 1 kinase protein was searched in the Protein Data Bank (http://www.rcsb.org). For the present work, the structure of the human TGF-beta receptor I kinase with inhibitor having PDB ID 1PY5 was selected [28]. The sequence length of chain A is 346. BIOVIA Discovery Studio (DS) 2019 was utilized to prepare the protein structure for subsequent steps by removing water atoms and heteroatoms.

## Docking studies of target macromolecule with metformin

The prepared protein structure was saved in Protein Data Bank file format. The receptor macromolecule, TGF-ß receptor 1 kinase protein was then opened in PyRx software [29]. The structure data file of metformin molecule was downloaded from PubChem. The structure data file (SDF) of metformin was designated as ligand. The docking score of TGF-ß receptor 1 kinase protein with metformin was determined using Autodock vina platform in PyRx software [29]. Docking was done to determine how well metformin binds to the target protein.

## Pharmacophore generation of metformin and TGFBR1 docked complex

At first, the docked complex of metformin with TGF-ß receptor 1 kinase was utilized to generate protein-based pharmacophore using the software, Ligandscout® 4.3 [30]. The generated structure-based pharmacophore was validated using active set and decoy set retrieved from

enhanced Database of Useful Decoys (DUDe) [31]. The active set contained 281 molecules while the decoy set comprised of 8674 molecules. Both these sets of molecules were collected from DUDe web-server. All of these compounds were allocated for TGFBR1 target macromolecule which werefound from subset section of DUDe server. The screening was conducted using the features- pharmacophore fit (scoring function), match all query features (screening mode) and get best matching conformation (retrieval mode).

## Pharmacophore mapping to screen natural compounds

All the features of the generated pharmacophore were used for screening purposes. A library of 29,000 natural compounds were retrieved from NPASS database [32]. The structures available in the compound library were downloaded in SDF format. The entire library was then converted to ldb file format. Afterwards, the generated pharmacophore was mapped against the natural compound library of 29,000 natural molecules collected from NPASS compound library [32]. Then compounds with same features of the pharmacophore were detected with the help of default settings.

## Molecular docking of leads with TGF-ß receptor 1 kinase

TBR1 (PDB ID: 1PY5) was cleaned using BIOVIA Discovery Studio 2019 by eliminating water atoms and other protein chains. Protein data bank (PDB) structure file was created for the protein's cleaned structure. PyRx was used to blindly dock the active substances against TGFBR1 in order to determine their binding affinities [29]. The receptor protein was opened using PyRx software [33]. The Structure Data Files (SDF) for 60 compounds were retrieved from PubChem. The chosen compounds' SDFs were used as inputs and were assigned the ligand label. Using the PyRx software's Autodock vina platform, TGFBR1 and 60 compounds were docked [33]. BIOVIA Discovery Studio 2019 was used to visualize the ligand-protein interaction. Lead compounds' affinities for binding were contrasted with those of the control ligands [34].

In the second part of docking analyses, top ten leads based on binding affinities found in blind docking operation were selected for site specific docking with TGFBR1. The interacting site of metformin with TGFBR1 was specified by selecting the amino acid residues with which metformin interacts. A grid was generated around the target site of metformin in TGFBR1 and the docking of ten leads were performed. This was done to comprehend the tendency of top leads to interact with target site of metformin in the macromolecular protein structure.

## ADMET prediction

ESOL Class, CYP inhibitor, BBB permeant, Lipinski rule, GI absorption and bioavailability grade using SWISSADME were used to determine whether natural substances were drug-like. The webtool pkCSM is used to forecast the toxicity parameters of the chosen compounds. The structure data files for ligands were collected from PubChem. The downloaded files were sequentially uploaded to the SwissADME [35] and pkCSM webservers [36]. Results from the generation were tabulated.

## Oral bioavailability estimation

The top 5 molecules' bioavailability radar profiles were collected from the SWISSADME website in order to determine drug-likeness of the compounds. These five chemicals' canonical smile IDs from the Pubchem database were initially gathered. Then, for each chemical, the canonical smile ID was uploaded to the SWISSADME website [35]. The website displayed

previously recorded details on the requested chemical. The radar images came from the bio-availability section [37].

## Molecular dynamic simulation studies

We performed 100 ns MD simulations by using 'Desmond v3.6 Program' in Schrödinger (Academic version) in Linux operating system to evaluate binding stability of metformin and selected candidate compounds. An orthorhombic periodic boundary box with specific volume and with a distance of 10 Å was assigned in the predetermined TIP3P water model which was used as the dynamics system. This solvated system was equilibrated with suitable ions $Na^+$ and $Cl^-$ with a concentration of 0.15 M. Minimization and relaxation of the system was performed using the default protocol of Desmond module using OPLS3 force field parameters. NPT ensemble maintaining 300 K and one atmospheric (1.01325 bar) pressure was used with recording intervals of 100 picoseconds. To analyze the results, Simulation Interaction Diagram (SID) of the Desmond module in the Schrödinger package has been used. The stability of the complex structure has been evaluated according to the root-mean-square deviation (RMSD) and root mean-square fluctuation (RMSF) that were generated by the analysis of the simulation trajectory.

## Estimation of biological activity using PASS

PASS (Prediction of Activity Spectra for Substances) is a tool for assessing an organic drug-like molecule's overall biological potential [38]. Based on the structure of organic substances, PASS makes simultaneous predictions of numerous different forms of biological activity. PASS can be utilized in order to predict the biological activity profiles of virtual molecules before their chemical synthesis and biological testing. PASS online web-server was used to predict anti-neoplastic activities of our top suggested compounds. The canonical SMILES of the compounds with CID 72473, 10316977, 45140078, 34755 were incorporated into the designated search box in the web-server. The biological activities associated with parameter of Pa (probability of activity) for antineoplastic feature were collected and tabulated.

## Results and discussion

### Transforming growth factor-beta (TGF-β) receptor type 1(TGFBR1) mRNA expression is upregulated in different cancers

A comprehensive online tool for examining cancer OMICS data is UALCAN [39]. UALCAN is developed to retrieve publicly accessible cancer OMICS data such as The Cancer Genomic Atlas (TCGA) more easily. TGFBR1 mRNA expression in the most prevalent cancer types (breast, lung, colon, and rectum cancer cells, as well as liver cancer cells) was examined using the UALCAN server. According to the Fig 2, TGFBR1 expression upregulated in the following cancers: CHOL-cholangiocarcinoma (bile duct cancer), COAD- colon adenocarcinoma (colon cancer), ESCA-Esophageal carcinoma (Esophageal carcinoma),GBM- glioblastoma multiforme (Brain cancer),HNSC-Head and neck squamous cell carcinoma (Head and neck cancer), LIHC- Liver hepatocellular carcinoma (Liver cancer),PCPG- Pheochromocytoma and Paraganglioma (neuroendocrine cancer), SARC-Sarcoma (bone cancer), THCA- Thyroid carcinoma (Thyroid cancer) and STAD- Stomach adenocarcinoma (Stomach adenocarcinoma). Among the most prevalent type of cancer, TGFBR1 expression in colon and rectum, stomach, and liver cancer was examined using the TCGA datasets. TGFBR1 was expressed more strongly in cancerous tissue than in normal tissue in each of these kinds of cancer. As illustrated in Fig 2, TGFBR1 expression in malignant tissue is indicated by the red box plot,

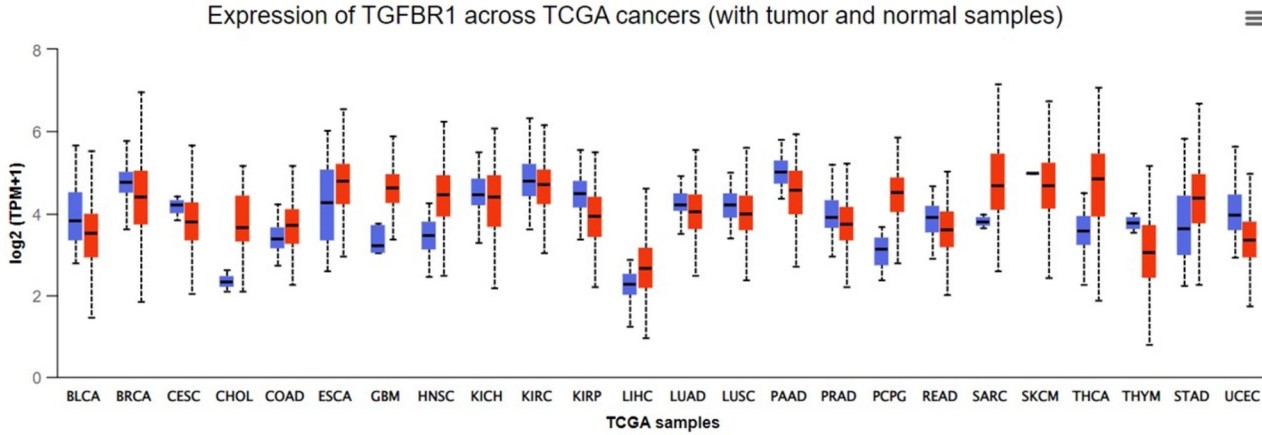

**Fig 2. Expression analysis of TGBR1 across multiple cancers.** Red box plots indicate cancerous tissue expressions vs blue box plots indicating tissue expression in normal tissue. The high overall median of the red box plots vs blue box plots indicate TGBR1 can be highly overexpressed in many cancer cell types.

whereas normal tissue expression is indicated by the blue box plot. We observed that the blue line graph is at a reduced level than the red line graph for all of the above-mentioned cancer kinds. This demonstrates that these tumors have high TGFBR1 expression levels. Thus, the protein can be targeted for development of anti-cancer medication due to its high expression in prevalent cancer type tissues.

In numerous cellular activities, the transforming growth factor-ß (TGF-ß) signaling pathway is crucial [15]. When cancer is advanced, TGF-ß shifts from acting as a tumor suppressor in healthy or dysplastic cells to acting as a tumor promoter [8]. The activation of Smad-independent pathways, along with the loss of TGF-ß's tumor-suppressive capabilities, is crucial for its pro-oncogenic functions [40]. In contrast, it is commonly accepted that the Smad-dependent pathway is essential in TGF-ß's tumor-suppressive functions [41]. It has been thought that TGF-signaling is an effective therapeutic target. Moreover, TR-I tumor-specific mutations have been identified in lymphoma, pancreatic, biliary, and breast cancer [42]. There is a lot of interest in creating TGF-ß signaling inhibitors for the treatment of cancer as a result of the revelation of TGF-ß's carcinogenic effects.

## Molecular docking of metformin with TGF beta receptor kinase 1

Metformin was taken as control ligand for TßR1 inhibition as it works as a suppressor of TGF-ß signalling and this signalling includes two receptors TßR1 and TßRII. Nearly all patients with newly diagnosed type II diabetes mellitus are prescribed the biguanide metformin, which is an oral anti-hyperglycemic medication of first choice [11]. Although metformin has been used to treat diabetes in Europe since 1957, its exact molecular mechanisms are still not entirely understood. Since the drug has been shown to have anti-cancer potentials, it can be utilized to develop anti-malignant medicines with better efficacy. Also, metformin has multi-protein targeting potentials as it is known to affect more than one pathway to regulate insulin sensitizing activities [43]. The multi-pathway targeting activities of metformin can yield fruitful results in developing drugs for cancer since the disease is known to act through various mechanisms. Here, we focused on the effect of metformin on TGFBR1 protein which is a major component of TGF-ß pathway.

The majority of TGF-ß signaling inhibition at the receptor kinase level is achieved by small-molecule inhibitors. These inhibitors typically work by attaching to the TGF-ß receptor (TGFBR) 1 kinase's ATP-binding domain and preventing its phosphorylation upon interaction with TGFBR2. This prevents TGFBR1 from activating downstream targets like Smad2 and Smad3 by maintaining TGFBR1 in a dormant configuration. Small-molecule inhibitors of Smad signaling have demonstrated to be sufficient to suppress tumor growth and tumor cell proliferation, decrease the advancement of cells into an epithelial-to-mesenchymal transition-like phenotype, suppress TGFß1 mediated transcriptional responses and suppress tumor cell migration and invasion [44].

Xiao et al., examined the impact of metformin on the generation of ligand-induced TRßII dimers, which is a result of TGF-ß ligand-receptor contact and necessary for receptor activation [20]. HeLa cells were treated with metformin and was observed using total internal reflection fluorescence microscopy. They found a dose-dependent inhibition of the percentage of dimers produced by TGF-ß1 when exposed to metformin. In the same study, they analyzed the effect of metformin on two of the components involved in the entire dimerization process which were TGF-ß molecule and TRßII receptor through computational tools such as molecular docking and molecular dynamics simulation. However, the effect of metformin on the third and most important component of the dimerization process that is, TGFBR1 is yet to be elucidated. Here, in this study we have looked into how metformin affects TGFBR1.

In another research, scientists showed that metformin causes inhibition of collagen synthesis by TGF-ß1 in cultured adult mouse cardiac fibroblasts [45]. Metformin also causes inhibition of pressure overload-induced generation of transforming growth factor (TGF)-ß1 in mice hearts. In CFs, metformin reduced Smad3's phosphorylation in response to TGF-ß1. Additionally, in CFs, metformin reduced Smad3's nuclear translocation and transcriptional activity. Thus, it can be assumed that metformin affects TGF-ß pathway. Li et al., demonstrated in their study that metformin attenuated activation of Transforming Growth Factor-β signaling as shown by reduced expression of pSmad2 and pSmad3. Dimerization is a prior step to Smad activation and so it can be inferred that metformin may influence this step which leads to reduced expression of phosphorylated Smads in subsequent steps [46]. Fig 3 illustrates the above mentioned phenomenon of dimerization inhibition of metformin in a concise manner.

The docking score of the control ligand, metformin (CID 4091) to TßR1 was determined in order to understand the binding energy of metformin towards TßR1.The more the negative docking score, the stronger would be the complex of ligand-protein complex. A single biomolecule's docking score is the potency of the binding relationship with its ligand or binding partner (e.g., drug or inhibitor). Binding energy is typically measured and reported using the equilibrium dissociation constant (KD), which is used to evaluate and rank order the energies of bimolecular interactions [47]. The KD value should be lower which indicates stronger docking score of the chemical for its target [48]. The KD value increases as the affinity and binding between the protein of interest and the ligand become weaker. Hydrogen bonds, electrostatic contacts, hydrophobic forces, and Van der Waals forces are examples of non-covalent intermolecular interactions that can impact binding energy [49]. The presence of extra molecules could affect the docking score between a ligand and its receptor [50]. A strong ligand-protein binding would disrupt the carcinogenetic mechanism of TßR1. Metformin had docking score of -4.7 kcal/mol where it showed hydrogen bond interaction with residue ASP351, LYS213, ASN338 and LYS337 and electrostatic bond with ASP351(Table 1).

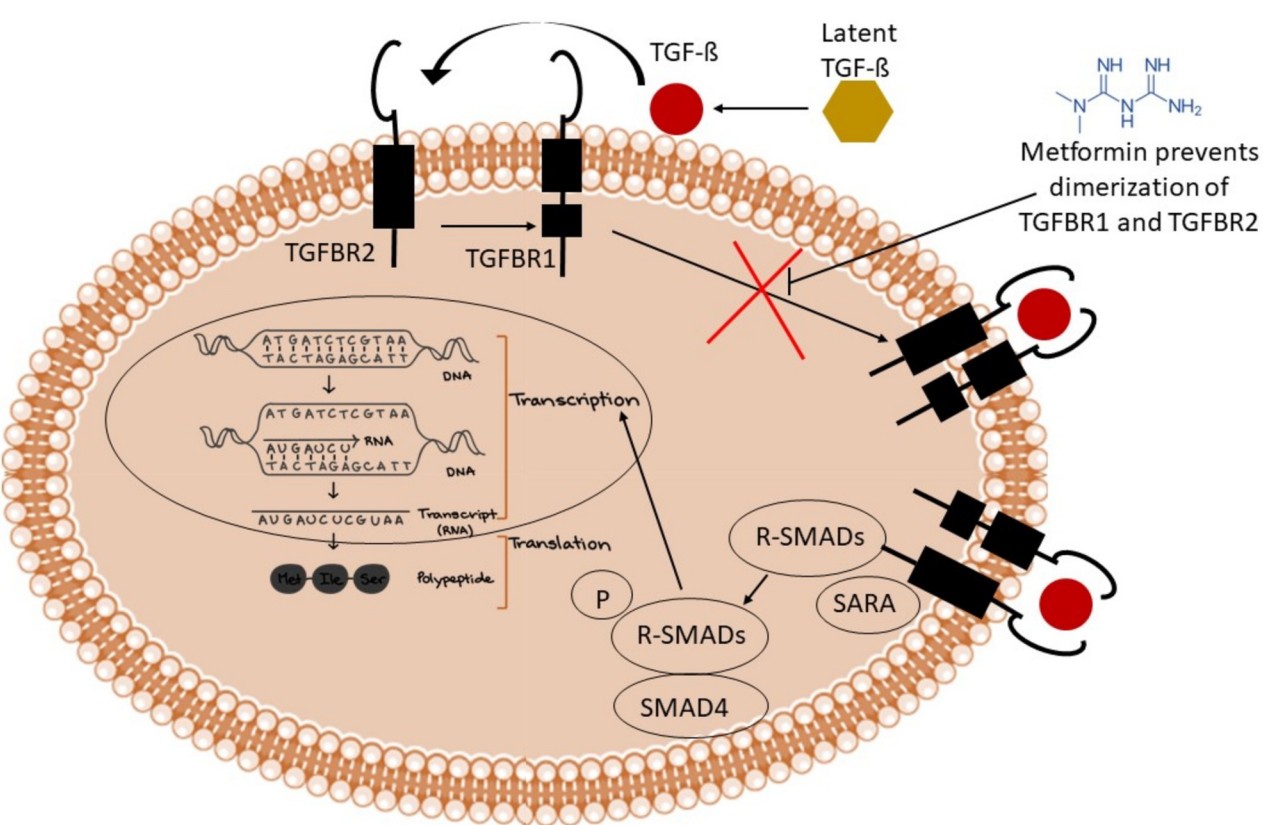

**Fig 3. Mechanism of inhibition of dimerization of TGFBR1 and TGFBR2 complex by metformin.** TGFß-I attaches to TßRII, luring TßRI into a heteromeric complex of TßRI-TßRII. TßRI is phosphorylated during the formation of this complex. This results in the phosphorylation of receptor-regulated Smads. The translocation and aggregation of R-Smad/co-Smad complexes in the nucleus subsequently regulate target genes. Metformin has been shown to inhibit TGF-ßI-induced EMT, which is necessary for the proliferation of cancer.

## Pharmacophore model generation with metformin-TGB1 docked complex and pharmacophore mapping

Assessment of metformin's interaction with Transforming Growth Factor-Beta receptor 1 protein has been carried out through molecular docking and subsequent analysis of interaction between the small molecule and target macromolecule. The docked complex was saved in Protein Data Bank File format. The file was opened in Ligandscout® software. It was used to generate structure based pharmacophore of metformin interacting with TGFB1 protein. The generated pharmacophore was validated before carrying out the screening process. The Receiver operating characteristic (ROC) curve generated from the validation process is given in S1 Fig. The curve is just above the dotted line which indicates that the pharmacophore is valid for screening purposes. The generated pharmacophore of metformin and TGFBR1 complex showed the following features: one Hydrogen bond donor and two positive ionisable areas with 3.19Å distance between these two areas (S2 Fig). Afterwards, we ran the pharmacophore against our collection of 29,000 natural chemicals from NPASS database using default settings in Ligandscout® software. We found candidate inhibitors that shared several pharmacophore characteristics with the combined pharmacophore and showed promising binding affinities towards the target macromolecule (Fig 4). There was a total of 60 compounds with a hit rate of 0.002069%.

**Table 1. Docking score and interaction between protein residues and ligand.**

| Ligand(Binding affinity) | Residue | Bond |
|---|---|---|
| CID 4091 (-4.7 kcal/mol) | ASP351 | Hydrogen |
| | LYS213 | Hydrogen |
| | ASN338 | Hydrogen |
| | LYS337 | Hydrogen |
| | ASP351 | Electrostatic |
| CID 44592809 (-11.4 kcal/mol) | ASN344 | Hydrogen |
| | ASP281 | Hydrogen |
| | HIS256 | Hydrogen |
| | GLU257 | Hydrogen |
| | LYS343 | Hydrophobic |
| | LYS342 | Hydrophobic |
| | TYR291 | Hydrophobic |
| | VAL341 | Hydrophobic |
| CID 72473 (-11 kcal/mol) | GLY286 | Hydrogen |
| | GLY345 | Hydrogen |
| | THR346 | Hydrogen |
| | CYS347 | Hydrogen |
| | CYS348 | Hydrogen |
| | VAL341 | Hydrophobic |
| | HIS283 | Hydrophobic |
| | ASN344 | Hydrophobic |
| | LYS343 | Hydrophobic |
| | CYS348 | Hydrophobic |
| | LYS342 | Hydrophobic |
| | LEU340 | Hydrophobic |
| CID 10316977 (-11 kcal/mol) | THR346 | Hydrogen |
| | GLU257 | Hydrogen |
| | CYS347 | Hydrogen |
| | VAL341 | Hydrophobic |
| | LYS342 | Hydrophobic |
| | LEU260 | Hydrophobic |
| | CYS348 | Hydrophobic |
| | HIS283 | Hydrophobic |
| | HIS285 | Hydrophobic |
| | TYR291 | Hydrophobic |
| CID 14162516 (-9.2 kcal/mol) | THR346 | Hydrogen |
| | CYS347 | Hydrogen |
| | ASP281 | Hydrogen |
| | GLU257 | Hydrogen |
| | VAL341 | Hydrophobic |
| | TYR291 | Hydrophobic |
| | LYS343 | Hydrophobic |
| | ASN344 | Hydrophobic |
| | ASP281 | Electrostatic |

(*Continued*)

**Table 1.** (Continued)

| Ligand(Binding affinity) | Residue | Bond |
|---|---|---|
| CID 15286763 (-8.9 kcal/mol) | CYS348 | Hydrogen |
| | HIS285 | Hydrogen |
| | TYR291 | Hydrophobic |
| | LYS343 | Hydrophobic |
| | ASN344 | Hydrophobic |
| | LEU340 | Hydrophobic |
| | CYS348 | Hydrophobic |
| | VAL341 | Hydrophobic |
| | LYS342 | Hydrophobic |
| CID 45140078 (-8.6 kcal/mol) | ASN344 | Hydrogen |
| | ASP281 | Hydrogen |
| | HIS256 | Hydrogen |
| | GLU257 | Hydrogen |
| | LYS343 | Hydrophobic |
| | LYS342 | Hydrophobic |
| | TYR291 | Hydrophobic |
| | VAL341 | Hydrophobic |
| CID 3081545 (-8.4 kcal/mol) | HIS256 | Hydrogen |
| | ASP281 | Hydrogen |
| | GLU257 | Hydrogen |
| | CYS347 | Hydrogen |
| | ASN344 | Hydrogen |
| | VAL341 | Hydrophobic |
| | TYR291 | Hydrophobic |
| | ASN344 | Hydrophobic |
| | LYS343 | Hydrophobic |
| | CYS348 | Hydrophobic |
| | LYS342 | Hydrophobic |
| CID 36294 (-8.3 kcal/mol) | ASN344 | Hydrogen |
| | THR346 | Hydrogen |
| | CYS347 | Hydrogen |
| | HIS283 | Hydrogen |
| | SER308 | Hydrogen |
| | LEU340 | Hydrophobic |
| | CYS348 | Hydrophobic |
| | HIS283 | Hydrophobic |
| CID 34755 (-8.3 kcal/mol) | ASP281 | Hydrogen |
| | HIS256 | Hydrogen |
| | ILE259 | Hydrogen |
| | HIS285 | Hydrogen |
| | CYS348 | Hydrogen |

## Molecular docking of leads with TGF beta receptor kinase 1

Metformin was chosen as the control ligand for TßR1 inhibition in molecular docking experiments because it suppresses TGF-ß dimerization, which involves the TßR1 and TßRII receptors. To comprehend the binding energy of metformin for TßR1, the docking score of the

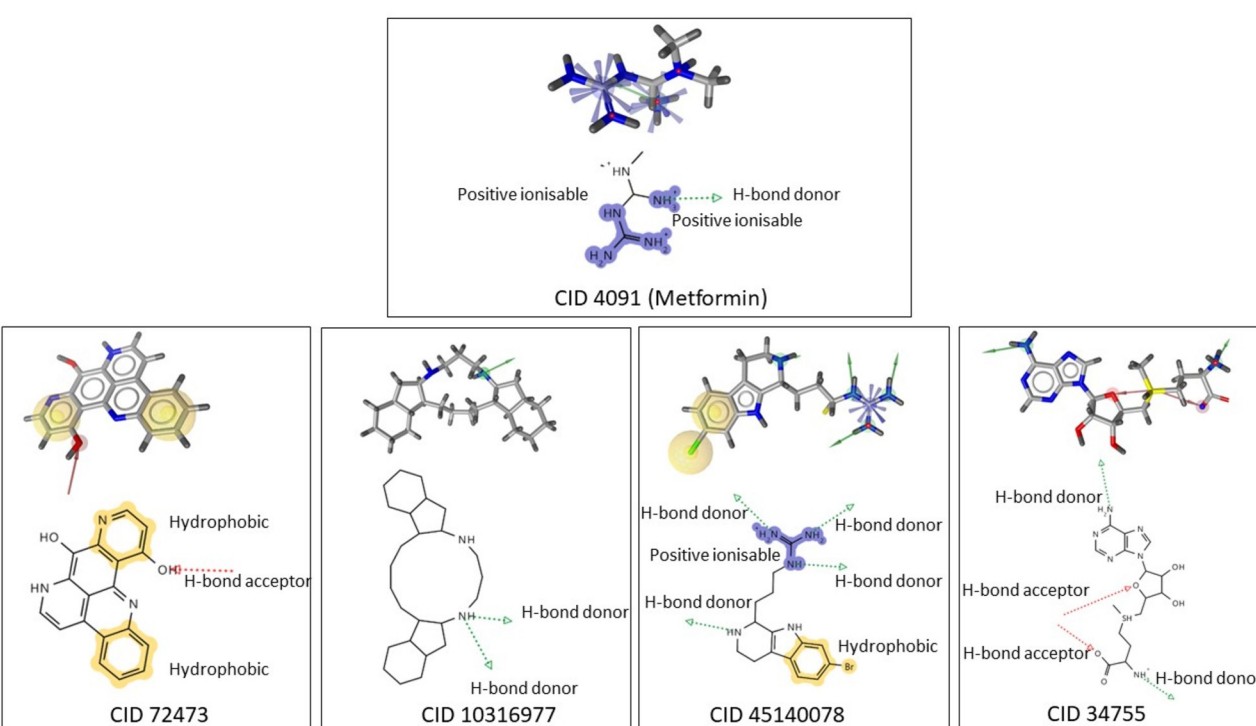

**Fig 4. Pharmacophore model of metfprmin-TGFBR1 complex and selected leads with CID 72473, 10316977, 45140078 and 34755 complexed with TGFBR1.**

control ligand metformin (CID 4091) to TßR1 was calculated. With a docking score of -4.7 kcal/mol, metformin interacted with the residues ASP351, LYS213, ASN338 and LYS337 via hydrogen bonds, and it also formed an electrostatic binding with ASP351 (Table 1).

Among 29,000 compounds from NPASS library, 60 compounds showed similar pharmacophore features. These 60 compounds were docked with the target macromolecule TßR1.The compounds which showed similarity in pharmacophore features showed greater docking score with TßR1. CID 44592809 showed the best binding ability with protein with a docking score -11.4 kcal/mol and it mostly formed hydrogen and hydrophobic bond with the protein. The selected ligands showed docking score in the range -11.4 to -8.3 kcal/mol (Table 1). The chemical structures of top ligands with good binding affinity scores are shown in S3 Fig.

The ligands showed hydrogen bond with protein in residues ASN344, ASP281, GLU257, THR346, HIS256 and other important sites of protein. The hydrophobic interaction also occurred at LYS342, LYS343, ASN344, VAL341 and TYR291.The detailed interaction of our finally selected top five compounds have been highlighted at Figs 5 and 6. Also, the interaction diagram of additional compounds has been depicted in S4 Fig.

The crystal structures of the type I TGF-β receptor (TRI or ALK5) solved in the presence or absence of FKBP12 have shown the structural underpinnings of the activation switch [51]. The GS domain adopts a helix-loop-helix folding motif that spans the apex of the kinase amino lobe β4 strand in both forms. The core GS loop sequence 185-TTSGSGSGLP-194 contains serine and threonine residues that are targeted by type II receptors. When the GS domain is dormant, FKBP12 lies atop it and protects the GS loop from the type II receptor. The downstream helix αGS2 serves as the binding site for it. Leu195 and Leu196, two residues from this helix,

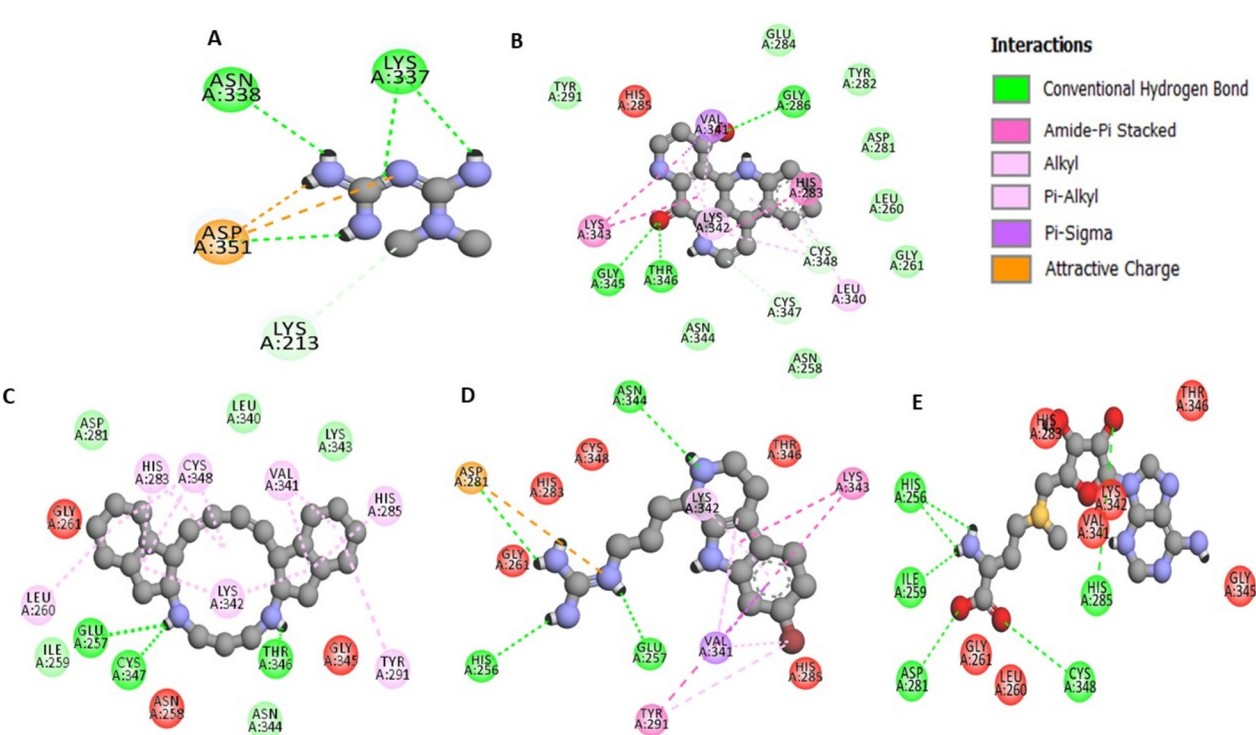

**Fig 5. Two-Dimensional Interaction diagram of metformin and lead compounds docked with TGFBR1 (A-CID 4091, B-CID 72473, C-CID 10316977,, D-CID 45140078, and E-CID 34755).**

are inserted into the FKBP12 macrolide-binding region and compete with rapamycin for binding.

The GS loop is pushed into the kinase domain by FKBP12 binding, where it creates an inhibitory wedge between the amino lobe β-sheet and the αC helix. By pulling the carboxy-terminal end of the αC helix out and swinging the amino terminus into the ATP pocket, this bends the kinase amino lobe. As a result, the catalytic salt bridge between TGFRBRI Asp245 (C helix) and Lys232 (β3 strand) is destroyed and the kinase domain adopts an inactive conformation. A variety of interactions between the arginine side chains, which are strictly conserved in type I receptors but divergent in type II receptors, stabilize the inhibitory complex. For instance, Arg203 connects the GS2 helix to the kinase domain, while Arg255, which is carboxy terminal to the αC helix holds the GS loop that is connected to it.

A salt bridge between Arg372 (activation loop) and the DLG motif further blocks the binding sites for ATP and substrate (Asp351) [51, 52]. In our analysis, it has been shown that metformin has a docking score of -4.7 kcal/mol with TGFBR1 protein. Metformin forms hydrogen bond and salt bridge with Asp351 and thus can be predicted to block the binding of ATP and substrate to the protein. Also, metformin forms one, two, and one conventional Hydrogen bond with Lys213, Lys 337 and Asn338 respectively.

In the second part of molecular docking analyses, site specific docking was carried out separately for top ten leads with TGFBR1 where the interacting amino acid residues of metformin-TGFBR1 complex were defined prior to docking operation. This was done to understand the affinities of leads towards the target site of metformin in TGFBR1. The second part of our analysis showed that the lead compounds with CID 10316977, CID 72473, CID 44592809, CID

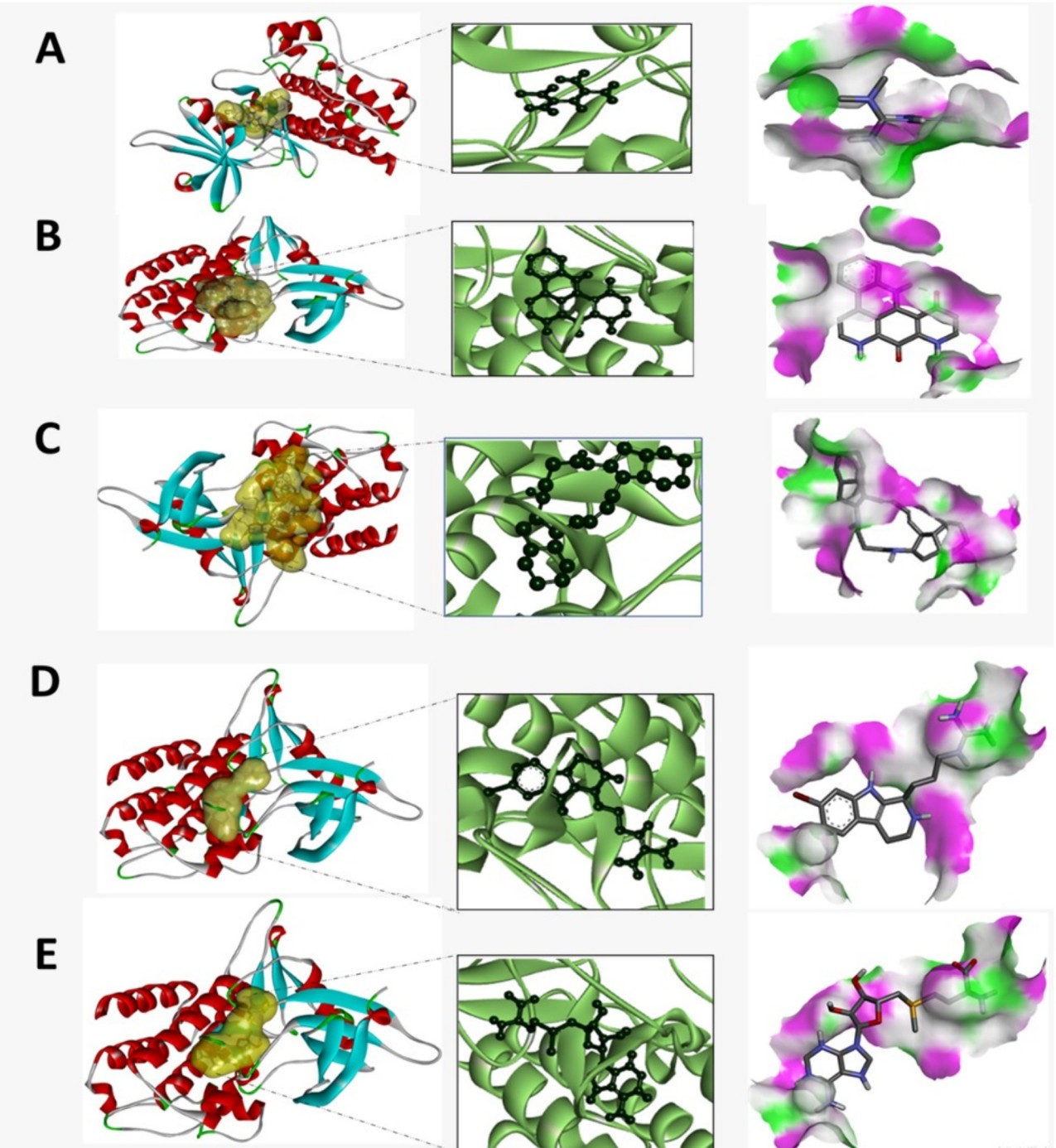

**Fig 6. Interaction of protein and ligand and displaying of hydrogen bond for A- CID 4091 B- 72473 C- CID 10316977 D- CID 45140078 E- CID 34755.**

45140078, CID 15286763, CID 14162516, CID 2724998, CID 3081545, CID 36294 and CID 34755 showed better binding affinities than metformin in site specific docking (S2 Table). The results show that the leads can interact with macromolecular target site of metformin effectively.

**Table 2. Admet properties of compounds and metformin (CID 4091).**

| Ligand | MW | iLOGP | ESOL Class | GI absorption | Lipinski #violations | Bioavailability Score | Synthetic Accessibility |
|---|---|---|---|---|---|---|---|
| 44592809 | 616.97 | 5.86 | Poorly soluble | High | 2 | 0.17 | 7.75 |
| 72473 | 299.28 | 1.55 | Soluble | High | 0 | 0.55 | 2.96 |
| 10316977 | 368.6 | 4.55 | Poorly soluble | High | 1 | 0.55 | 5.99 |
| 14162516 | 271.36 | 1.5 | Soluble | High | 0 | 0.55 | 3.08 |
| 15286763 | 243.35 | 2 | Soluble | High | 0 | 0.55 | 3.23 |
| 45140078 | 350.26 | 1.88 | Soluble | High | 0 | 0.55 | 3.18 |
| 3081545 | 539.58 | 1.92 | Highly soluble | Low | 3 | 0.17 | 6.8 |
| 36294 | 467.51 | 0.58 | Highly soluble | Low | 2 | 0.17 | 6.42 |
| 34755 | 398.44 | 0 | Highly soluble | Low | 1 | 0.55 | 4.9 |
| 2724998 | 212.29 | 2.27 | Soluble | High | 0 | 0.55 | 1.98 |
| 4091 | 129.16 | 0.34 | Highly Soluble | High | 0 | 0.55 | 3.02 |

## ADMET analysis of compounds

The top compounds discovered through docking analysis were examined for ADME (absorption, distribution, metabolism, and excretion) profiling. Before small biomolecules can be further developed into medications that may be used inside of human bodies at useful doses, it is crucial to take into account the ADME parameters. In Table 2, various ADME parameters are listed that includes molecular weight, iLOGP, ESOL class, GI absorption, Lipinski violations, bioavailability score and synthetic accessibility.

The SWISSADME website was used to identify the ADME (absorption, distribution, metabolism, and excretion) profiles of the natural compounds and metformin. The Lipinski filter was used to identify the active ingredient's druglikeness characteristics. Besides other parameters like MW, iLogP, ESOL Class, GI absorption, bioavailability score, and synthetic accessibility parameters were retrieved from the server. These criteria were used to determine the ADME profile of the compounds [53]. According to Table 2, the compounds 72473, 10316977, 14162516, 1528676, 45140078, 3081545. 34755,2724998 showed good GI absorption, and bioavailability scores suggesting good ADME properties. These compounds also didn't violate Lipinski Rule over the threshold of 1 which is admissible and suggests lead potential of the compounds. The synthetic accessibility scores of the compounds are also low, suggesting good lab reproducibility. Based on this data, literature review and previous docking results we selected 72473,10316977, 45140078 and 34755 for further oral bioavailability and molecular dynamics studies.

The pkCSM webserver was used to illustrate the toxicity profile of selected phytochemicals and metformin (Table 3). We analyzed the maximum tolerated dose in human, oral rat acute

**Table 3. Toxicity analysis from pkCSM.**

| Ligand | Max. tolerated dose (human) (log mg/kg/day) | Oral Rat Acute Toxicity (LD50) (mol/kg) | Oral Rat Chronic Toxicity (LOAEL) (log mg/kg_bw/day) | T.Pyriformis toxicity (log ug/L) | Minnow toxicity (log mM) |
|---|---|---|---|---|---|
| Metformin | 0.902 | 2.453 | 2.158 | 0.25 | 3.972 |
| CID 72473 | -0.067 | 2.791 | 0.923 | 0.285 | 2.162 |
| CID 10316977 | -0.082 | 2.582 | 0.643 | 0.299 | 0.19 |
| CID 45140078 | 0.259 | 2.672 | 2.535 | 0.288 | 0.588 |
| CID 34755 | 0.463 | 2.482 | 2.019 | 0.285 | 3.178 |

toxicity (LD50), oral rat chronic toxicity (LOAEL), T.Pyriformis toxicity and minnow toxicity. Compounds with CID 72473, CID 10316977, CID 45140078, and CID 34755 showed less maximum tolerated dose than metformin. Oral rat acute toxicity (LD50) of CID 72473, CID 10316977, CID 45140078, and CID 34755 are more than metformin. T.Pyriformis toxicity and minnow toxicity of the compounds CID 72473, CID 10316977, CID 45140078, and CID 34755 occur at higher dose than metformin. Compounds with CID 45140078, CID 34755 and metformin have oral rat chronic toxicity around 2.

## Prediction of oral bioavailability

SWISSADME was used to gather data on the top drugs' oral bioavailability metrics. We retrieved bioavailability radar illustrations using the SMILE IDs ((retrieved from the PubChem database) of the top chemicals in the SWISSADME webserver.

The Bioavailability Radar makes it possible to quickly assess the drug likeliness of a compound. The pink area represents the ideal range for each property, including Lipophilicity (XLOGP3 between − 0.7 and + 5.0), Flexibility (no more than 9 rotatable bonds), Size (MW between 150 and 500 g/mol), Polarity (TPSA between 20 and 130 angstrom), In-solubility (log S not higher than 6), and In-saturation (sp³ hybridized fraction of carbons not less than 0.25).

In Fig 7 we can see that the radar characteristics of metformin and 45140078 are almost similar suggesting bioavailability properties just like metformin. Compounds 10316977, 72473, 34755 also show good distribution suggesting good bioavailability properties.

## Molecular dynamic simulation studies to analyze stability of metformin and leads interacting with TGF beta receptor kinase 1

The dynamic behavior and stability of protein-ligand complexes can be simulated and assessed by molecular dynamics simulations [54]. Therefore, the docked metformin and filtered compounds complexes with TBR1 were subjected to molecular dynamics simulation of 100ns to determine their stability and characteristics in a simulated environment. RMSD (root mean square deviation) values of protein ligand complex are an indication of whether simulation of the protein-ligand complex is stable within a reference structure [55]. Fig 8 shows the RMSD

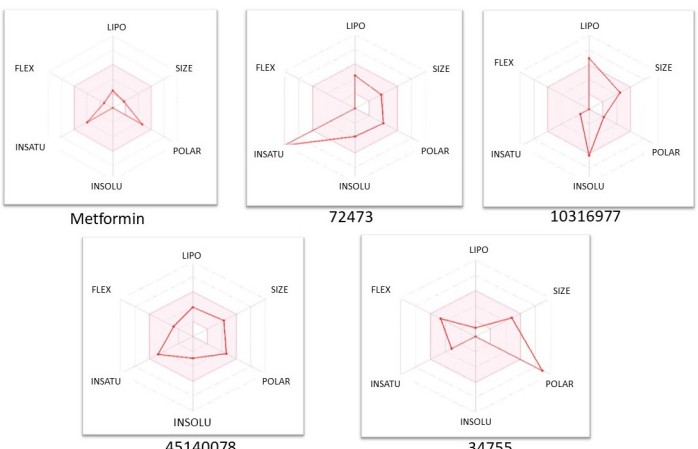

**Fig 7. Bioavailability radar images of top five compounds.** The almost identical radar properties of 45140078 and metformin suggest that the latter has similar bioavailability features to the former. The distribution of compounds 10316977, 72473, and 34755 also suggests that they have strong bioavailability qualities.

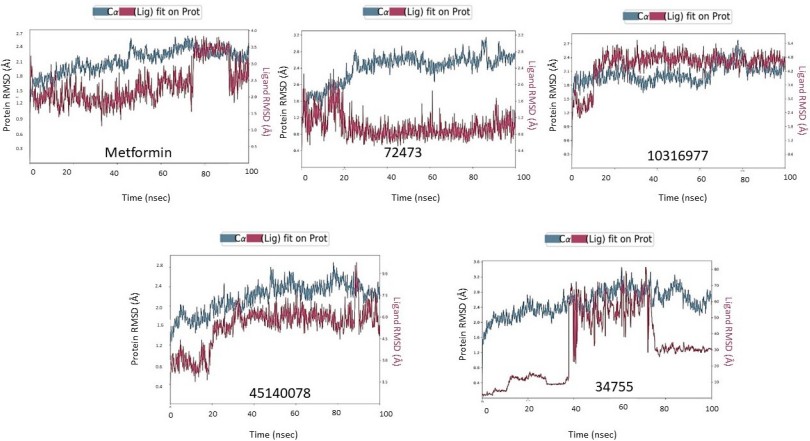

**Fig 8. Trajectory plots of molecular dynamics (MD) of target protein-drug complex.** Protein ligand-root mean square deviation (PL-RMSD) with respect to time (nanoseconds) during MD simulation of TBR1 complexed with Metformin, 72473, 10316977, 45140078And 34755.

of TBR1 protein on left Y-axis, while the Y-axis on right shows the ligand RMSD profile aligned on protein backbone of the studied complexes. The fluctuations of RMSD value between 0.1 and 0.3 nm are deemed stable. This indicates a short-ranged protein conformational change [50].

The RMSD plot in Fig 8 indicates that the metformin-TGF-beta receptor-1 kinase (TBR1) (PDB ID) complex stabilized shortly after commencing the simulation around 1.5 ns. Then until 74ns, it remained around the 2–3 Å mark with a deviation of about 1 Å. During 74 nanoseconds we see a high deviation around 1.8 Å (from 1.558–3.237 Å) which is still within the acceptable range of < 3 Å. The complex hovered around the 3.5–3 Å mark with a fluctuation of about 1 Å until 91ns and then returned to the initial 2–3 Å mark and remained stable with fluctuations about 1Å. The protein Cα backbone of the TBR1 protein remained stable throughout the whole simulation with fluctuations around 1–1.5Å s. Thus, the overall RMSD plot suggests that the metformin is stably bound to TBR1 binding site and has not diffused away from the bound position. In the case of RMSD plot of 72473-TBR1 complex (Fig 8) both the ligand and Cα backbone showed fluctuations of around only 1–1.5 Å indicating high stability and tight binding of the complex throughout the whole 100ns simulation.

RMSD plot of 10316977-TBR1(Fig 8) complex shows stabilization around 10.3ns at the 4-5Å mark and remained stable with around 1–1.5 Å fluctuations until the end of simulation. The Cα back bone also remained stable with around 1–1.5 Å throughout the whole simulation suggesting a stable complex between the ligand and protein.

RMSD data 45140078-TBR1(Fig 8) showed stabilization at time 19.4 second and continued fluctuating with an admissible deviation of 1–3 Å around the 4–7 Å mark until 88ns. At 88ns, it shows a fluctuation of around 4.2ns (From 5.576–9.78). This spike lasted only for one nanosecond and the overall trajectory then again showed fluctuations < 3 Å and remained stable until the end of simulation. The protein backbone was also stable throughout the simulation. So, it can be said that overall 45140078-TBR1 formed a stable complex. From Fig 5 it can be seen that 34755-TBR1 is the most unstable complex with fluctuations ranging from 10–60 Å. The ligand diffused away from the protein and its trajectory showed frantic movement in the simulated environment. This might be due to no pure bonds present and only a hydrophobic pocket binding the ligand and protein together.

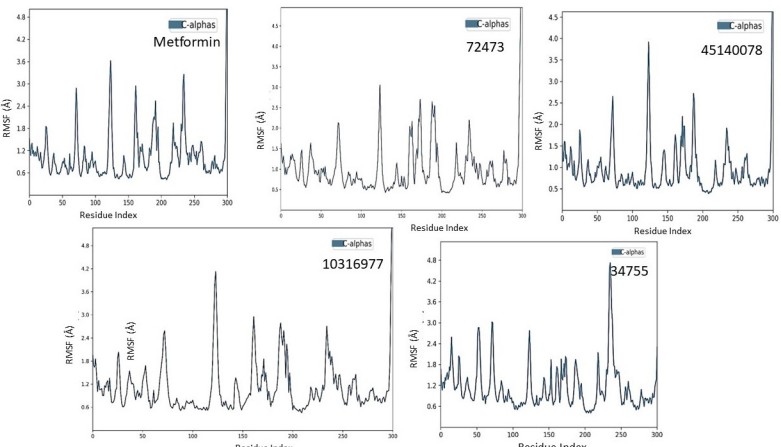

**Fig 9. Trajectory plots of molecular dynamics (MD) of target protein-drug complex.** Protein-root mean square fluctuation (P-RMSF) with respect to time (nanoseconds) during MD simulation of TBR1 complexed with Metformin, 72473, 10316977, 45140078 and 34755.

In molecular dynamics simulation, it is important to evaluate whether the protein chain undergoes any local conformational change which can be analyzed using the Root Mean Square Fluctuation (RMSF) [56]. The RMSF (root-mean-square fluctuations) were evaluated and plotted to compare each residue's flexibility in the protein-ligand complexes. The protein-RMSF showed minimal fluctuation for all complexes throughout the 100 ns simulation. The RMSF did not deviate much during the simulation period, and the average RMSF values were found similar for all the complexes and is presented in Fig 9.

Fig 10 shows the residue interaction diagram of TBR1 with metformin and selected compounds. Interactions that last more than 30% of the simulation time are highlighted and considered. Fig 8 shows residue-ligand interactions in the form of stacked bar-charts during the 100ns simulation. The stacked bar charts signify the time and type of interactions over the course of the trajectory. In general, four types of receptor-ligand interactions are recognized. Hydrophobic, Ionic, and Hydrogen Bonds and water Bridges are highlighted using different colors in the chart. The y axis represents a residue fraction versus residues at the x axis. A residue fraction of 0.5 means for 50% of the simulation time the contact is maintained. The residue interaction diagram with the interactions bar-chart gives an in-depth view of the interactions taking place during the simulation.

From the interaction diagram of metformin (Fig 10) we can see that metformin forms hydrogen bond and ionic interaction (charged positive) with As351 residue for the longest (over 50%) time frame. For this dual interaction, As351 residue interaction index is over 1 and stands at 2.5. It also interacts with Lys335 via salt bridge through chlorine for around 40% of the simulation time. From the bar chart we can also see that Lys213 of TGFBR1 forms H-bonds and water bridges with metformin as the 3$^{rd}$ most interactive residue with a interaction fraction of over 0.5 (Fig 10).

72473 forms strong hydrogen-bond interactions with Thr-346 for over 80% of the duration of simulation via hydroxyl and pyridinium group of its structure and is the most prominent interaction seen during the simulation which accounts for the high stability of the complex (Fig 7). It also forms strong water bridges with residues Val-341 and Lys-343 and strong hydrophobic interactions with Lys342 and Leu-260 (Fig 10).

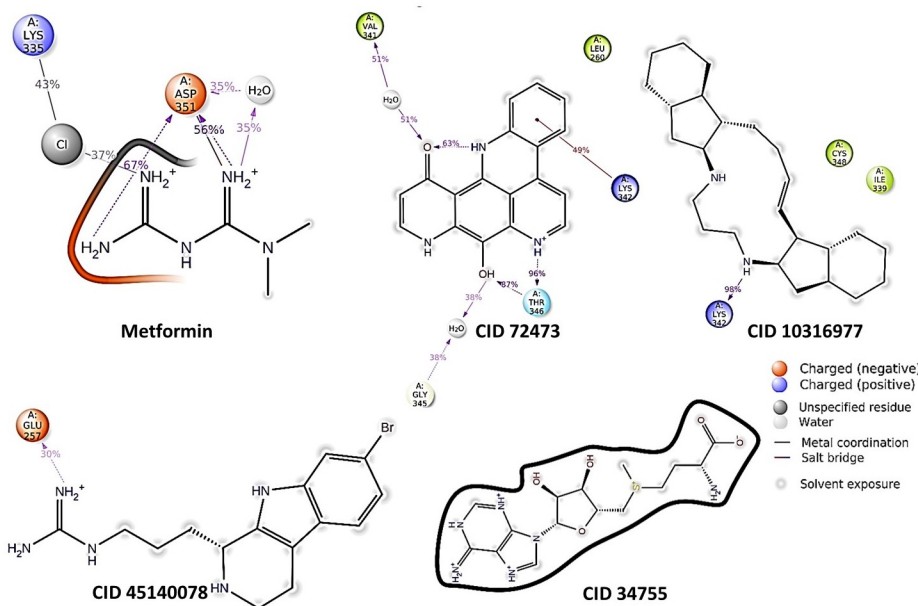

**Fig 10. Interaction plots of molecular dynamics (MD) of target protein-drug complex.** Interaction diagram during MD simulation of TBR1 complexed with metformin, 72473, 10316977,45140078 and 34755. Color labels at the end signify the interaction taking place for each column.

Like 73473, compound 10316977 also forms interactions with Lys342 but this interaction is a hydrogen bond and is maintained throughout the whole period of the simulation (Fig 10). This is due to this strong hydrogen bond interaction that the complex was found to be stable. It also forms strong hydrophobic interactions with Cys342, Ile339 and Leu305 residues (Fig 11).

45140078 forms hydrogen bonds with many residues such as Glu257, Ile259, Leu-254 and Asp281. But this H-bonds have low residency and the strongest among them is that with Glu-257 with an interactive fraction of 0.7. It also forms strong hydrophobic Lys-342 bond (Fig 11).

Compound 34755 does not form any H-bonds, ionic, hydrophobic or water bridge interactions to impart the complex any stability. That is why erratic behavior was found in the trajectory of the compound during the whole time of simulation (Figs 10 and 11).

## Biological activities of lead compounds

The biological activities of the compounds with CID 72473, 10316977, 45140078, 34755 were analyzed using PASS online computational tool. Meperidine (CID 72473) showed probability of activity (Pa) greater than 0.5 for the most number of anti-cancer activities such as Antineoplastic, Antineoplastic (colorectal cancer), Antineoplastic (colon cancer), Antineoplastic (lung cancer), Antineoplastic (breast cancer), Antineoplastic alkaloid, Antineoplastic (glioma), Prostate cancer treatment and Antineoplastic (bladder cancer). Compound with CID 10316977 showed Pa score of 0.528 for Antineoplastic (squamous cell carcinoma). The molecule CID 34755 has a probability of activity score 0.598 for the biological activity antineoplastic-antimetabolite. For molecule 45140078, the biological activity, antineoplastic (endocrine cancer) showed a Pa score of 0.252. The result of biological activities of lead compounds collected from PASS online is presented in S3 Table.

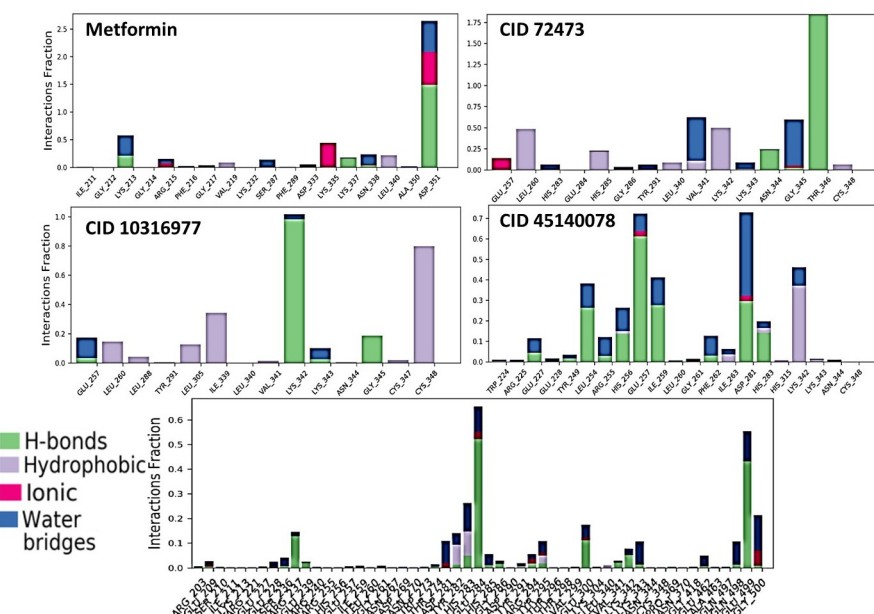

**Fig 11. Normalized stacked interaction bar charts during MD simulation of TBR1 complexed with metformin, 72473, 10316977,45140078 and 34755.** Color labels at the end signify the interaction taking place for each column.

## Conclusion

Cancer patients may be benefited from therapies targeting specific macromolecules that play important roles in malignancies. The main aim of the current study was to find novel compounds with metformin-like activities against TGFBR1. The pharmacophore that metformin forms with TGFBR1 signifies the interactions of the drug with the protein. These interactions are necessary for the action of metformin on TGFBR1. Structure based pharmacophore mapping of 29,000 phytochemicals filtered out lead compounds. The initial pharmacophore mapping filtered out 60 compounds. Molecular docking was performed in order to further filter these compounds,. Then, ADMET analysis was done. The filtering parameters brought out top compounds with potentials to be developed as drugs, with nearly identical binding affinity and stability to metformin. Molecular dynamics simulation was carried out in the final stage to understand how the suggested compounds interact with TGFBR1 under human body conditions. The compounds with CID 72473, 10316977 and 45140078 showed promising binding affinities and formed stable complexes during dynamics simulation. Future investigations can include enzymatic assays to investigate biological activities of leads, determination of IC$_{50}$ values and cytotoxicity studies on cancer cell lines. The results of the study may be useful for future investigations and the development of potential anticancer medications.

## Supporting information

**S1 Fig. The ROC curve generated from Ligandscout software for validation of pharmacophore.**
(TIF)

**S2 Fig. Features of pharmacophore generated for metformin-TGFBR1 protein complex.**
(TIF)

**S3 Fig. Structures of top 10 leads with highest binding affinities with TGFBR1 protein.**
(TIF)

**S4 Fig. Two-dimensional interaction diagram of metformin and lead compounds docked with TGFBR1 for compounds with CID 14162516, CID 15286763, CID 3081545, CID 36294 and CID 44592809.**
(TIF)

**S1 Table. Binding affinities of 60 lead compounds with TGFBR1 protein.**
(XLSX)

**S2 Table. Binding affinities of top ten leads with TGFBR1 using site specific docking.**
(XLSX)

**S3 Table. Probability of activity score retrieved from PASS online web-server.**
(XLSX)

## Author Contributions

**Conceptualization:** Rumman Reza, Niaz Morshed, Md. Nazmus Samdani.

**Data curation:** Niaz Morshed, Md. Nazmus Samdani.

**Formal analysis:** Md. Selim Reza.

**Investigation:** Rumman Reza, Md. Nazmus Samdani.

**Methodology:** Rumman Reza, Niaz Morshed, Md. Nazmus Samdani.

**Resources:** Md. Selim Reza.

**Software:** Rumman Reza, Niaz Morshed, Md. Nazmus Samdani.

**Supervision:** Md. Selim Reza.

**Visualization:** Md. Nazmus Samdani.

**Writing – original draft:** Rumman Reza, Niaz Morshed, Md. Nazmus Samdani.

**Writing – review & editing:** Rumman Reza, Md. Selim Reza.

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
