## [Decision Letter · Decision Letter 0]

29 Mar 2023

PONE-D-23-07740Identification of metformin-like phytochemicals targeting transforming growth factor (TGF)-beta receptor I kinase for development of anti-cancer medicines: A computational analysisPLOS ONE

Dear Dr. Reza,

Thank you for submitting your manuscript to PLOS ONE. After careful consideration, we feel that it has merit but does not fully meet PLOS ONE’s publication criteria as it currently stands. Therefore, we invite you to submit a revised version of the manuscript that addresses the points raised during the review process. Please submit your revised manuscript by May 13 2023 11:59PM. If you will need more time than this to complete your revisions, please reply to this message or contact the journal office at plosone@plos.org. Please include the following items when submitting your revised manuscript:A rebuttal letter that responds to each point raised by the academic editor and reviewer(s). You should upload this letter as a separate file labeled 'Response to Reviewers'.A marked-up copy of your manuscript that highlights changes made to the original version. You should upload this as a separate file labeled 'Revised Manuscript with Track Changes'.An unmarked version of your revised paper without tracked changes. You should upload this as a separate file labeled 'Manuscript'.

We look forward to receiving your revised manuscript.

Kind regards,

Ahmed A. Al-Karmalawy, Ph.D.

Academic Editor

PLOS ONE

Journal Requirements:

3. We note that you have stated that you will provide repository information for your data at acceptance. Should your manuscript be accepted for publication, we will hold it until you provide the relevant accession numbers or DOIs necessary to access your data. If you wish to make changes to your Data Availability statement, please describe these changes in your cover letter and we will update your Data Availability statement to reflect the information you provide

Reviewers' comments:

Reviewer's Responses to Questions

**Comments to the Author**

1. Is the manuscript technically sound, and do the data support the conclusions?

Reviewer #1: Partly

Reviewer #2: Partly

2. Has the statistical analysis been performed appropriately and rigorously? 

Reviewer #1: Yes

Reviewer #2: N/A

3. Have the authors made all data underlying the findings in their manuscript fully available?

Reviewer #1: Yes

Reviewer #2: Yes

4. Is the manuscript presented in an intelligible fashion and written in standard English?

Reviewer #1: No

Reviewer #2: No

5. Review Comments to the Author

Reviewer #1: The work exerted is so appreciated. However, some points need to be addressed before acceptance. So, a major revision may be required to improve the manuscript.

1) Some typos and grammatical errors need to be corrected.

2) The manuscript need to be revised extensively by native English people.

3) Although, TGF was defined in the title but it was not defined in the abstract. Please define it in the abstract part for the first time only.

4) Some phytochemicals that previously reported to act on TGF-β should be included in the introduction part.

5) The study was conducted on a library of 29,000 compounds which can be treated as Hit compounds. So, the selected 60 compounds could be considered as lead compounds NOT Hit compounds. Please correct through the whole manuscript.

6) Please cite references for SWISSADME and PKCSM web tools

7) ADME properties prediction should include a reference for comparison

8) Heat map and MM-GBSA should be conducted in molecular dynamics part

9) SMAD sometimes written capitalized and sometimes written in small letters. Please unify

10) Since in silico studies are not enough tools to prove a mechanisms of action, so i highly recommend to conduct even one biological test on the most relevant candidate to prove the paper perspective.

11) Conclusion part have to be improved.

Reviewer #2: The manuscript entitled "Identification of metformin-like phytochemicals targeting transforming growth factor (TGF)-beta receptor I kinase for development of anti-cancer medicines: A computational analysis” is an attention grabbing topic however, significant major revisions are required. My observations/comments are given below.

1- The title should be reformulated.

2- The filtration system used in this work was based on generation of the pharmacophore model. However, no presence for the generated pharmacophore model inside the manuscript showing essential features, dimensions, distances, and angles.

3- Validation of the pharmacophore model should be addressed properly.

4- You should also supply images for the most fitted compound with the pharmacophore model.

5- In vitro cytotoxic activities of the most active compound is highly recommended.

6- Enzymatic assay should be carried out, if possible, against transforming growth factor (TGF)-beta receptor I kinase for at least the most active member.

7- Flexible alignment between metformin and the most active members may be used as an additional filtration system.

8- The conclusion part should be rephrased.

9- In the molecular docking part, it was noticed that the binding mode of metformin is completely different from the hit compounds! How can you explain this point.

10- The chemical structures of the hit compounds should be added as supporting information.

11- All grammatical errors and the typo mistakes should be corrected in the whole manuscript.

6. PLOS authors have the option to publish the peer review history of their article (what does this mean?). If published, this will include your full peer review and any attached files.

Reviewer #1: No

Reviewer #2: No

---

## [Author Response · Author response to Decision Letter 0]

5 May 2023

Ref: [PONE-D-23-07740] - [EMID:05405f898ee519b8]

Journal: PLOS ONE

Date: May, 2023

To

Ahmed A. Al-Karmalawy, Ph.D.

Academic Editor

PLOS ONE

Dear Dr. Ahmed A. Al-Karmalawy, 

Thank you for your consideration of our manuscript [PONE-D-23-07740] - [EMID:05405f898ee519b8].

We thank all the reviewers for their critical assessment of our initial submission. The concerns raised by each reviewer were helpful in improving overall quality of our work. A summary of the major and minor changes made to the manuscript in accordance to the reviewers’ comments have been appended below. Here, we have addressed all the reviewers’ concerns point by point. We have assessed all the issues in the reviewers’ comments and made suitable changes accordingly. Also, the manuscript has been formatted according to the journal formatting guidelines of PLOS ONE.

We have thoroughly revised the manuscript and believe our current manuscript represents a significant improvement over our initial submission. English language of the revised manuscript has been thoroughly scrutinized by our research assistants who have expertise in writing manuscripts for reputed journals. We can assure you that the quality of English language used in the updated manuscript now meets the expected standards. 

We believe the manuscript is now suitable for publication in your reputed journal.

Kind Regards,

Md. Selim Reza, PhD

Professor

Department of Pharmaceutical Technology, 

Faculty of Pharmacy, 

University of Dhaka, 

Dhaka-1000, Bangladesh

Email: selimreza@du.ac.bd

Major and minor corrections made in accordance to comments from Reviewer-1:

Reviewer #1: The work exerted is so appreciated. However, some points need to be addressed before acceptance. So, a major revision may be required to improve the manuscript.

1) Some typos and grammatical errors need to be corrected.

We have made major and minor changes in sentence structure. Grammatical errors have been critically assessed and removed. The revised manuscript is an improvised version in terms of sentence structure and grammatical construction.

2) The manuscript need to be revised extensively by native English people.

To the best of our knowledge, we have made appropriate changes in sentence structure and grammatical construction. The corrections include-

• Simplification of complex sentences

• Correction of grammatical errors

• Incorporation of synonymous words to avoid unnecessary repetition

• Addition of linking words in between sentences

• Homogeneity in tense usage of sentences

• Inclusion of comprehensive sentence structure throughout the manuscript

The overall quality of English Language has been improved markedly in the revised version of our manuscript. English language of the revised manuscript has been thoroughly scrutinized by our research assistants who have expertise in writing manuscripts for reputed journals. We can assure you that the quality of English language used in the updated manuscript now meets the expected standards.

3) Although, TGF was defined in the title but it was not defined in the abstract. Please define it in the abstract part for the first time only.

TGF has been defined in the abstract in the revised version of the manuscript. The abstract of the manuscript has been modified. We believe that the current abstract matches with findings of this research work. It adequately presents the significance of the research done and its impact for future research work in this field.

4) Some phytochemicals that previously reported to act on TGF-β should be included in the introduction part.

Some of the small molecule inhibitors reported to act on TGF-β has been included in the introduction part. Several new review and research papers have been cited in the revised version of the manuscript. The flow of writing is improved and the newly added information is in accordance with scientifically proven findings.

5) The study was conducted on a library of 29,000 compounds which can be treated as Hit compounds. So, the selected 60 compounds could be considered as lead compounds NOT Hit compounds. Please correct through the whole manuscript.

The raised concern has been addressed carefully by considering the 60 compounds as lead compounds. 

6) Please cite references for SWISSADME and PKCSM web tools

Proper citations of the referred articles have been incorporated in accordance to the style followed by this journal

7) ADME properties prediction should include a reference for comparison

Metformin is a well-established drug. The ADMET properties of metformin has been used as a point of reference for the lead compounds.

8) Heat map and MM-GBSA should be conducted in molecular dynamics part

We used the academic version of DESMOND software. The academic version of the software does not contain plug in for performing MM-GBSA calculations. 

9) SMAD sometimes written capitalized and sometimes written in small letters. Please unify

“Smad” has been addressed in small letters throughout the entire manuscript. 

10) Since in silico studies are not enough tools to prove a mechanisms of action, so i highly recommend to conduct even one biological test on the most relevant candidate to prove the paper perspective.

The experimental design used in the present study reflects the current trend in computer-aided drug design-based experimentation. The design employed has been effective in several other in-silico based studies for finding out effective inhibitor of druggable targets. A software program called PASS (Prediction of Activity Spectra for Substances) was created as a tool for assessing an organic drug-like molecule's overall biological potential. Based on the structure of organic substances, PASS makes simultaneous predictions of numerous different forms of biological activity. In order to predict the biological activity profiles of virtual molecules before their chemical synthesis and biological testing, PASS can be utilized. 

We used PASS online web-server to predict anti-neoplastic activities of our top suggested compounds.

11) Conclusion part have to be improved.

The revised version includes a comprehensive and well-organized conclusion section. The results are correctly presented in the most understandable way so that the readers may appreciate the importance of the study.

Major and minor corrections made in accordance to comments from Reviewer-2:

Reviewer #2: The manuscript entitled "Identification of metformin-like phytochemicals targeting transforming growth factor (TGF)-beta receptor I kinase for development of anti-cancer medicines: A computational analysis” is an attention grabbing topic however, significant major revisions are required. My observations/comments are given below.

1- The title should be reformulated.

The title has been changed to “Search for potential anti-cancer phytochemicals with metformin-like activities against transforming growth factor (TGF)-beta receptor I kinase: Pharmacophore mapping, molecular docking and molecular dynamics simulation analysis”.

The current title focus on

• The target protein

• Identification of compounds with anti-cancer effects

• Screening of natural phytochemicals

• Approaches used in the methodology

2- The filtration system used in this work was based on generation of the pharmacophore model. However, no presence for the generated pharmacophore model inside the manuscript showing essential features, dimensions, distances, and angles.

Pharmacophore model of metfprmin-TGFBR1 complex has been represented in figure 4 in the updated manuscript. Also, the essential features and dimensions are indicated in supplementary figure 2 as well as in the result section. 

3- Validation of the pharmacophore model should be addressed properly.

In the updated manuscript, validation of the pharmacophore model has been addressed in the methodology section, “Pharmacophore generation of metformin and TGFBR1 docked complex”. Also, Receiver operating characteristic (ROC) curve is supplied in supplementary figure S1. 

4- You should also supply images for the most fitted compound with the pharmacophore model

Pharmacophore model of metfprmin-TGFBR1 complex and leads has been represented in figure 4 in the updated manuscript.

5- In vitro cytotoxic activities of the most active compound is highly recommended.

The entire study follows a computational drug discovery approach that has been adopted to identify most probable leads. The suggested compounds have high probability to work against TGFBR1 protein. However, we could not include in vitro assays in the manuscript. Future analyses can include laboratory based assays. 

6- Enzymatic assay should be carried out, if possible, against transforming growth factor (TGF)-beta receptor I kinase for at least the most active member.

To find the most likely leads, a computational drug discovery approach has been used throughout the entire investigation. The likelihood that the indicated chemicals will inhibit the TGFBR1 protein is very high. Enzymatic assays, however, were not able to be included in the publication. Future analysis could incorporate tests conducted in a lab.

7- Flexible alignment between metformin and the most active members may be used as an additional filtration system.

We used several filtering parameters, such as:

• Pharmacophore alignment score

• Docking score

• ADMET score

• PASS score

• Molecular dynamic simulation

These parameters thoroughly selected the compound most suitable for anti-cancer activity.

8- The conclusion part should be rephrased.

The conclusion part has been rephrased.

9- In the molecular docking part, it was noticed that the binding mode of metformin is completely different from the hit compounds! How can you explain this point?

We have a added an additional docking analysis in the updated manuscript. In the second part of docking analyses, top ten leads based on binding affinities found in blind docking operation were selected for site specific docking with TGFBR1. The interacting site of metformin with TGFBR1 was specified by selecting the amino acid residues with which metformin interacts. A grid was generated around the target site of metformin in TGFBR1 and the docking of ten leads were performed. This was done to comprehend the tendency of top leads to interact with target site of metformin in the macromolecular protein structure. The binding affinity of leads in site specific docking is greater than that of metformin (supplementary table S2).

10- The chemical structures of the hit compounds should be added as supporting information.

Chemical structures of hit compounds are generated using ChemDraw software and included in supplementary information as figure S3. 

11- All grammatical errors and the typo mistakes should be corrected in the whole manuscript.

Some of the things that we have done to improve the logical flow of the manuscript include-

• Information on a particular topic has been clustered together into a single paragraph.

• Further details have been added in each section. The revised manuscript contains a well-structured and detailed discussion section.

• The findings are presented appropriately in the most simplified manner possible so that the readers understand the significance of the research done.

• Linking phrases have been utilized appropriately to maintaining the flow of writing.

---

## [Decision Letter · Decision Letter 1]

18 May 2023

PONE-D-23-07740R1Search for potential anti-cancer phytochemicals with metformin-like activities against transforming growth factor (TGF)-beta receptor I kinase: Pharmacophore mapping, molecular docking and molecular dynamics simulation analysisPLOS ONE

Dear Dr. Reza,

Thank you for submitting your manuscript to PLOS ONE. After careful consideration, we feel that it has merit but does not fully meet PLOS ONE’s publication criteria as it currently stands. Therefore, we invite you to submit a revised version of the manuscript that addresses the points raised during the review process.

The Reviewers raised some issues regarding the lack of* in vitro* studies for your work. First, I ask you to change the title and abstract to be clearer that the current study is a pure computational one. Second, you should try to satisfy the reviewers' comments as possible. Please submit your revised manuscript by Jul 02 2023 11:59PM. If you will need more time than this to complete your revisions, please reply to this message or contact the journal office at plosone@plos.org. Please include the following items when submitting your revised manuscript:A rebuttal letter that responds to each point raised by the academic editor and reviewer(s). You should upload this letter as a separate file labeled 'Response to Reviewers'.A marked-up copy of your manuscript that highlights changes made to the original version. You should upload this as a separate file labeled 'Revised Manuscript with Track Changes'.An unmarked version of your revised paper without tracked changes. You should upload this as a separate file labeled 'Manuscript'.

We look forward to receiving your revised manuscript.

Kind regards,

Ahmed A. Al-Karmalawy, Ph.D.

Academic Editor

PLOS ONE

**Reviewers' comments:**

Reviewer's Responses to Questions

**Comments to the Author**

1. If the authors have adequately addressed your comments raised in a previous round of review and you feel that this manuscript is now acceptable for publication, you may indicate that here to bypass the “Comments to the Author” section, enter your conflict of interest statement in the “Confidential to Editor” section, and submit your "Accept" recommendation.

Reviewer #1: All comments have been addressed

Reviewer #2: All comments have been addressed

Reviewer #3: (No Response)

2. Is the manuscript technically sound, and do the data support the conclusions?

Reviewer #1: Partly

Reviewer #2: Partly

Reviewer #3: Yes

3. Has the statistical analysis been performed appropriately and rigorously? 

Reviewer #1: Yes

Reviewer #2: N/A

Reviewer #3: Yes

4. Have the authors made all data underlying the findings in their manuscript fully available?

Reviewer #1: No

Reviewer #2: Yes

Reviewer #3: Yes

5. Is the manuscript presented in an intelligible fashion and written in standard English?

Reviewer #1: Yes

Reviewer #2: No

Reviewer #3: Yes

6. Review Comments to the Author

Reviewer #1: Although most points were addressed by the authors, but the enzymatic assay (Biological tests) has not been conducted yet. In my point of view, in silico studies ONLY are not enough tools. So, the paper cannot be accepted in its current form.

Reviewer #2: The authors have refuted most of my comments and therefore I recommend accepting the manuscript after considering two main points.. Firstly, the title of the manuscript should be changed to be shorter and clearer than that.. Secondly, the manuscript should undergo careful linguistic revision especially in the formulation of some sentences and grammatical rules.

Reviewer #3: To gain a better understanding of how metformin affects the TGF beta receptor 1 kinase at a molecular level, the authors conducted molecular docking experiments. Through these experiments and subsequent molecular dynamics simulations, it was predicted that metformin interacts with the transforming growth factor (TGF)-beta receptor I kinase. Additionally, a pharmacophore was created for the complex formed between metformin and TGF-ßR1, with the aim of identifying new compounds that possess similar pharmacophore features as metformin but with enhanced anti-cancer properties. The pharmacophore was used to conduct virtual screening of 29,000 natural compounds from the NPASS database using Ligandscout software. The screening identified 60 lead compounds that showed potential interaction with the metformin-TGF-ßR1 complex. These compounds were subjected to molecular docking, molecular dynamics simulations for 100 ns, and ADMET analysis. Among the tested compounds, CID 72473, 10316977, and 45140078 exhibited favorable binding affinities, formed stable complexes during dynamics simulations with the aforementioned protein, and thus hold promise for further development as anti-cancer medications. Some recommendations to be taken under consideration before publication of this manuscript:

1. In the pdf, some figures were found to be represented in low resolution. Please provide higher-resolution Figures in a separate file and assure providing them to the editorial office.

2. In the conclusion section, and since no invitro assays, I recommend authors to add two or more sentences about what kind of future biological assays could be performed to validate and follow up the computational-based findings.

3. In Line 226, (...was equilibrated with suitable ions Na+ and Cl-). Charges should be superscripted. Please check all over the manuscript.

4. Some language and editing errors have been found and a native English editing service is suggested before final acceptance of the manuscript.

5. In the final paragraph of the introduction, the workflow Fig. 1 should be explained in details within the text.

7. PLOS authors have the option to publish the peer review history of their article (what does this mean?). If published, this will include your full peer review and any attached files.

Reviewer #1: No

Reviewer #2: No

Reviewer #3: **Yes**

---

## [Author Response · Author response to Decision Letter 1]

17 Jun 2023

Ref: [PONE-D-23-07740] - [EMID:05405f898ee519b8]

Journal: PLOS ONE

Date: 02 June, 2023

To

Ahmed A. Al-Karmalawy, Ph.D.

Academic Editor

PLOS ONE

Dear Dr. Ahmed A. Al-Karmalawy, 

Thank you for your consideration of our manuscript [PONE-D-23-07740] - [EMID:05405f898ee519b8].

We appreciate the feedback from every reviewer on our amended submission. Each reviewer's critiques helped us to improve the overall caliber of our work. Below is a list of the major and minor edits made to the manuscript in response to the reviewers' comments. Here, we have addressed each issue raised by the reviewers in detail. All of the issues raised in the reviewers' comments have been evaluated, and the necessary improvements have been made. The manuscript has also been structured in accordance with PLOS ONE's journal formatting standards.

We have thoroughly revised the manuscript and believe our current manuscript represents a significant improvement over our previous submission. English language of the revised manuscript has been thoroughly scrutinized by our research assistants who have expertise in writing manuscripts for reputed international journals. We can assure you that the quality of English language used in the updated manuscript now meets the expected standards. 

We believe the manuscript is now suitable for publication in your reputed journal.

Kind Regards,

Md. Selim Reza, PhD

Professor

Department of Pharmaceutical Technology, 

Faculty of Pharmacy, 

University of Dhaka, 

Dhaka-1000, Bangladesh

Email: selimreza@du.ac.bd

Major and minor corrections made in accordance to comments from Reviewer-1:

Reviewer #1: Although most points were addressed by the authors, but the enzymatic assay (Biological tests) has not been conducted yet. In my point of view, in silico studies ONLY are not enough tools. So, the paper cannot be accepted in its current form.

We tried to include the enzymatic assay. But unfortunately, the required laboratory facilities to conduct this biological test was not available within the span of this research project.

However, increasing number of papers use computational methodology to screen large number of molecules that can save resources in a time-efficient manner. The findings of our research enlisted the most promising compounds from a library of 29,000 natural molecules. Future research on the suggested compound can focus on biological assays to provide a different dimension and aid in discovering anti-cancer medicaments.

Reviewer #2: The authors have refuted most of my comments and therefore I recommend accepting the manuscript after considering two main points.. Firstly, the title of the manuscript should be changed to be shorter and clearer than that.. Secondly, the manuscript should undergo careful linguistic revision especially in the formulation of some sentences and grammatical rules.

Firstly, the title of the manuscript has been changed to ‘Pharmacophore mapping approach to find anti-cancer phytochemicals with metformin-like activities against transforming growth factor (TGF)-beta receptor I kinase: An in silico study’.

Secondly, to the best of our knowledge, we have made appropriate changes in sentence structure and grammatical construction. The corrections include-

• Simplification of complex sentences

• Correction of grammatical errors

• Incorporation of synonymous words to avoid unnecessary repetition

• Addition of linking words in between sentences

• Homogeneity in tense usage of sentences

• Inclusion of comprehensive sentence structure throughout the manuscript

The revised version of our manuscript has a noticeably higher level of overall English language quality. Our research assistants, who have experience producing articles for reputable journals, have carefully reviewed the amended manuscript's English language. We can tell you that the modified document now complies with the anticipated criteria for English language usage.

Reviewer #3: To gain a better understanding of how metformin affects the TGF beta receptor 1 kinase at a molecular level, the authors conducted molecular docking experiments. Through these experiments and subsequent molecular dynamics simulations, it was predicted that metformin interacts with the transforming growth factor (TGF)-beta receptor I kinase. Additionally, a pharmacophore was created for the complex formed between metformin and TGF-ßR1, with the aim of identifying new compounds that possess similar pharmacophore features as metformin but with enhanced anti-cancer properties. The pharmacophore was used to conduct virtual screening of 29,000 natural compounds from the NPASS database using Ligandscout software. The screening identified 60 lead compounds that showed potential interaction with the metformin-TGF-ßR1 complex. These compounds were subjected to molecular docking, molecular dynamics simulations for 100 ns, and ADMET analysis. Among the tested compounds, CID 72473, 10316977, and 45140078 exhibited favorable binding affinities, formed stable complexes during dynamics simulations with the aforementioned protein, and thus hold promise for further development as anti-cancer medications. Some recommendations to be taken under consideration before publication of this manuscript:

1. In the pdf, some figures were found to be represented in low resolution. Please provide higher-resolution Figures in a separate file and assure providing them to the editorial office.

The fig 5 has been divided into fig5 and S4_fig by generating clearer image. Fig10 also have been updated with higher resolution. The TIFF file of all the images have been provided.

2. In the conclusion section, and since no invitro assays, I recommend authors to add two or more sentences about what kind of future biological assays could be performed to validate and follow up the computational-based findings.

The revised version includes a comprehensive and well-organized conclusion section. The results are correctly presented in the most understandable way so that the readers may appreciate the importance of the study.

The future assays that can be conducted has been appended to the conclusion section.

3. In Line 226, (...was equilibrated with suitable ions Na+ and Cl-). Charges should be superscripted. Please check all over the manuscript.

The charges have been superscripted in line 226 and all over the manuscript.

4. Some language and editing errors have been found and a native English editing service is suggested before final acceptance of the manuscript.

We made considerable revisions to the manuscript, and we think that it is now noticeably better than what we originally submitted. Our research assistants, who are skilled at writing articles for reputable international journals, have given the amended manuscript's English language an in-depth review. We can assure you that the modified manuscript's use of English now complies with all necessary criteria. 

5. In the final paragraph of the introduction, the workflow Fig. 1 should be explained in details within the text.

In the revised manuscript, workflow fig.1 has been explained in details in line no 122-124

---

## [Decision Letter · Decision Letter 2]

21 Jun 2023

Pharmacophore mapping approach to find anti-cancer phytochemicals with metformin-like activities against transforming growth factor (TGF)-beta receptor I kinase: An in silico study

PONE-D-23-07740R2

Dear Dr. Md. Selim Reza,

We’re pleased to inform you that your manuscript has been judged scientifically suitable for publication and will be formally accepted for publication once it meets all outstanding technical requirements.

Kind regards,

Ahmed A. Al-Karmalawy, Ph.D.

Academic Editor

PLOS ONE

Reviewers' comments:

Reviewer's Responses to Questions

**Comments to the Author**

1. If the authors have adequately addressed your comments raised in a previous round of review and you feel that this manuscript is now acceptable for publication, you may indicate that here to bypass the “Comments to the Author” section, enter your conflict of interest statement in the “Confidential to Editor” section, and submit your "Accept" recommendation.

Reviewer #1: All comments have been addressed

Reviewer #3: All comments have been addressed

2. Is the manuscript technically sound, and do the data support the conclusions?

Reviewer #1: Partly

Reviewer #3: Yes

3. Has the statistical analysis been performed appropriately and rigorously? 

Reviewer #1: I Don't Know

Reviewer #3: I Don't Know

4. Have the authors made all data underlying the findings in their manuscript fully available?

Reviewer #1: Yes

Reviewer #3: Yes

5. Is the manuscript presented in an intelligible fashion and written in standard English?

Reviewer #1: Yes

Reviewer #3: Yes

6. Review Comments to the Author

Reviewer #1: The paper entitled "Pharmacophore mapping approach to find anti cancer phytochemical.....'can be accepted in its current form

Reviewer #3: There are no additional remarks to be made; hence, it is appropriate to grant approval for the paper.

7. PLOS authors have the option to publish the peer review history of their article (what does this mean?). If published, this will include your full peer review and any attached files.

Reviewer #1: No

Reviewer #3: **Yes**

---

## [Editor Report · Acceptance letter]

5 Jul 2023

PONE-D-23-07740R2 

Pharmacophore mapping approach to find anti-cancer phytochemicals with metformin-like activities against transforming growth factor (TGF)-beta receptor I kinase: An *in silico* study 

Dear Dr. Reza:

I'm pleased to inform you that your manuscript has been deemed suitable for publication in PLOS ONE. Congratulations! Your manuscript is now with our production department. 

Kind regards, 

on behalf of

Dr. Ahmed A. Al-Karmalawy 

Academic Editor

PLOS ONE